# Returning CP-Observables to The Frames They Belong

Jona Ackerschott[1], Rahool Kumar Barman[2], Dorival Gonçalves[2],
Theo Heimel[1], and Tilman Plehn[1]

**1** Institut für Theoretische Physik, Universität Heidelberg, Germany
**2** Department of Physics, Oklahoma State University, Stillwater, USA

August 2, 2023

## Abstract

**Optimal kinematic observables are often defined in specific frames and then approximated at the reconstruction level. We show how multi-dimensional unfolding methods allow us to reconstruct these observables in their proper rest frame and in a probabilistically faithful way. We illustrate our approach with a measurement of a CP-phase in the top Yukawa coupling. Our method makes use of key advantages of generative unfolding, but as a constructed observable it fits into standard LHC analysis frameworks.**

# 1 Introduction

With the LHC continuing its success story of precision hadron collider physics, the size and complexity of the datasets of the upcoming Run 3 and HL-LHC are challenging the existing analysis methodology [1–3]. At the same time, the goal of LHC physics has moved from model-based searches for physics beyond the Standard Model (SM) to a comprehensive analysis of all its data, based on consistent analysis frameworks like the Standard Model effective theory [4–6].

The first step in any global analysis based on the fundamental principles of QFT is to determine the underlying symmetries, which are required to construct the effective Lagrangian. The, arguably, most interesting symmetry in the SM is $CP$, linked to cosmology through the Sakharov conditions for baryogenesis [7], and potentially realized in an extended Higgs sector [8]. In the language of effective theory, $CP$-violation in the Higgs coupling to vector bosons is loop-suppressed and arises at dimension six [9–13]. In contrast, $CP$-violation in Higgs couplings to fermions can appear at dimension four [14], making a $CP$-phase in the top Yukawa coupling the most sensitive link between baryogenesis and LHC physics [15–32].

Obviously, we do not want to leave the test of fundamental Lagrangian symmetries to a global analysis [33, 34] with limited control over experimental and theoretical uncertainties [35–39], including systematics from parton densities [40]. Instead, we should use dedicated (optimal) observables to target one fundamental symmetry at a time [12, 29, 30, 41–43]. In the Higgs-gauge sector, the optimal observable is the azimuthal angle between the two tagging jets in weak boson fusion. For associated top–Higgs production, the azimuthal angle between a charged lepton from one top decay and the down quark from the other plays a similar role. Accurately extracting it faces the challenge of identifying the corresponding decay jet. Another powerful observable probing the Higgs-top interaction is the Collins–Soper angle [23, 28, 29, 44]. Again, the challenge is to map it onto the observed final state after particle decays, parton shower, and detector effects.

Both of these observables illustrate the common problem that an optimal or ideal kinematic correlation is usually not defined on the reconstructed final state. So while an optimal observable provides full sensitivity without the need to consider additional phase space correlations, we pay a prize in its reconstruction.

The standard inference approach for such kinematic correlations is to approximate them at the reconstruction level. For this approximation, we can use a directly observable correlation at the reconstruction level or rely on some kind of algorithm. The approximation is unlikely to be optimal. An improved approach would be to encode the observable in a learned mapping, for instance, through neural networks. Fundamentally different and, in principle, optimal alternatives are simulation-based inference [45, 46] or the matrix element method [47–50], but they come at a significant numerical cost and are hard to re-interpret for other measurements.

For cases where an optimal observable is defined in some kinematic frame, we propose a simplified unfolding approach, where we unfold the reconstruction-level events to the appropriate reference frame, and then construct the optimal observable for the down-stream task. Unfolding or reconstructing events beyond the immediately available detector output is a long-standing problem [51–54], undergoing transformative progress through modern machine learning (ML) [55–63]. One key observation is that forward and backward simulations are completely symmetric when we interpret them as sampling from conditional probabilities [64, 65]. This motivates ML-unfolding through probabilistic inverse simulations [59, 60, 62], which allows us to reconstruct observables or parameters defined at any level of our forward simulations, for instance, unfolding detector effects, parton shower, particle decays [61],

all the way to measuring fundamental parameters [66].

This generative unfolding technique allows us to just reconstruct key observables, which have the advantage that they can be used in the standard analysis frameworks of ATLAS and CMS, but with a performance increase from the full set of kinematic correlations learned through the unfolding. To guarantee stable network predictions and to be able to quantitatively extract the training-induced network uncertainties, we use the Bayesian version [67] of the conditional normalizing flows [68, 69], for which the likelihood losses should lead to well-calibrated results. Eventually, this kind of analysis can serve as a simple starting point for ML-unfolding, as it can be expanded through additional observables step by step.

In this paper, we use $CP$-violation through a complex top Yukawa coupling to show how ML-unfolding techniques can construct and numerically encode observables in the reference frame where they are defined. In Sec. 2, we first describe our neural network architecture, the physics task, and the treatment of phase space. In Sec. 3, we introduce our reference process and discuss our results and potential generalization errors. Finally, Sec. 4 is reserved for summary and outlook.

## 2 Reconstructing observables by unfolding

In this study, we propose to use statistical unfolding through inverse simulation [59, 60] to construct kinematic observables in a specific partonic reference frame. While we are making use of unfolding techniques in constructing a given observable, the precision, control, and model dependence of the unfolding is not a limiting factor for our analysis. Instead, we treat the so-defined observable like any other kinematics construction.

### 2.1 Generative unfolding

Generative unfolding is based on the observation that a forward simulation from a parton-level event $x_{\text{part}}$ to a reco-level event $x_{\text{reco}}$ just samples from an encoded conditional probability,

$$r \sim \mathcal{N}(r) \xrightarrow{x_{\text{part}}} x_{\text{reco}} \sim p(x_{\text{reco}}|x_{\text{part}}) \qquad \text{(forward)} . \tag{1}$$

This simulation can be trivially inverted on the same training data, so we can unfold detector effects, initial-state jet radiation, or particle decays, by sampling from the inverse conditional probability,

$$r \sim \mathcal{N}(r) \xrightarrow{x_{\text{reco}}} x_{\text{part}} \sim p(x_{\text{part}}|x_{\text{reco}}) \qquad \text{(inverse)} . \tag{2}$$

In both cases, the standard training relies on paired events $\{x_{\text{part}}, x_{\text{reco}}\}$. Obviously, this training dataset leads to model dependence, which can be reduced by using iterative methods [62]. The target phase space of the inverse simulation or unfolding can be chosen flexibly, just unfolding detector effects [60, 62], but also jet radiation [60], particle decays [61], or sampling right into model parameter space using setups like BayesFlow [66]. Inference through conditional normalizing flows is standard in many fields of physics [70, 71].

Our generative network encoding the conditional probability defined in Eq. (2) is a conditional normalizing flow, specifically a conditional invertible neural network (cINN) [70], trained with a likelihood loss to guarantee a statistically correct and calibrated output. To link a batch of $B$ phase space points $x_i$ to a Gaussian latent space $r_i$ with the condition $c_i$, the

likelihood loss reads

$$\mathcal{L}_{\text{cINN}} = \sum_{i=1}^{B} \left( \frac{r_i(x_i; c_i)^2}{2} - \log \left| \frac{\partial\, r_i(x_i; c_i)}{\partial\, x_i} \right| \right). \tag{3}$$

## 2.2 Periodic splines

The main part of our cINN is built from coupling layers, specifically rational quadratic spline blocks [72], each followed by a random permutation. As we will discuss in Sec. 2.3, some phase space directions are periodic and lead to undesired boundary effects when we use these spline transformations. To understand this problem in detail, let us consider a spline transformation

$$g_\theta: \quad [-L, L] \to [-L, L], \tag{4}$$

with parameters $\theta$. This transformation is given by $K$ different monotonic rational quadratics, parameterized by $K + 1$ knot points $(x_k, y_k)$ and $K + 1$ derivatives. The boundaries are $(x_0, y_0) = (-L, -L)$ and $(x_K, y_K) = (L, L)$.

For periodic inputs the conditions $g(x_0) = y_0 = -L$ and $g(x_K) = y_K = L$ are unnecessarily restrictive and do not allow the network to map a distribution onto or past the boundaries, to represent points on a circle. In addition, we want $g'(x_0) = g'(x_K)$ for periodic inputs, which is not necessarily true [73]. The first issue can be fixed by replacing $g_\theta$ with

$$\tilde{g}_\theta(x) = g_\theta(x) + g_0 + 2Lk, \tag{5}$$

with an integer $k$ chosen such that $\tilde{g}(x)$ always lies within $[-L, L]$, and a new parameter $g_0$ added to $\theta$. To solve the second issue, we simply remove one of the derivative parameters from $\theta$ and set $\tilde{g}'_\theta(x_K) = \tilde{g}'_\theta(x_0)$. The resulting transformation $\tilde{g}_\theta$ is visualized in Fig. 1.

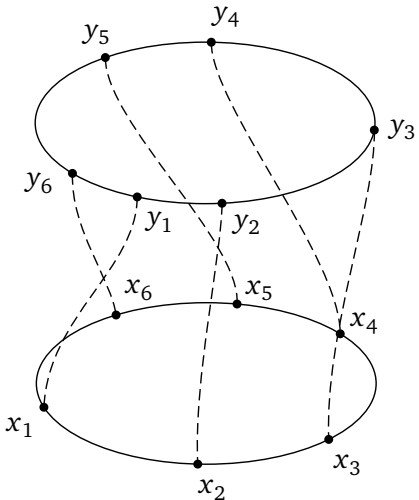

Figure 1: Visualization of a modified coupling transformation for periodic inputs. This transformation maps $K$ points $x_i$ to $K$ points $y_i$ on a circle, while we use rational quadratics to interpolate between two points. The modifications ensure that the $x_i$ and $y_i$ can be arbitrarily skewed in relation to one another, and that the derivative at $x_1$ is consistent with both adjacent rational quadratics.

| Parameter | Value |
|---|---|
| Block type | periodic rational quadratic spline blocks |
| Number of bins | 10 |
| Block Period | $2\pi$ |
| Block Domain (non-Periodic) | $[-5.0, 5.0] \rightarrow [-5.0, 5.0]$ |
| Number of Blocks | 16 |
| Layers per Block | 5 |
| Units per Layer | 256 |
| Weight Prior Type | Gaussian |
| Weight Prior $\log(\sigma^2)$ | 1.0 |
| Number of Epochs (Bayesian) | 100 (200) |
| Batch Size | 1024 |
| Optimizer | ADAM |
| Learning Rate | $2.0 \times 10^{-4}$ |
| Total number of training events | $\sim$1.2M |
| Training/Testing split | 80%/20% |

Table 1: Setup and hyper-parameters of the Unfolding-cINN.

With these modifications, the number of parameters encoding the transformation does not change. This means that we can use the same sub-network to determine $\theta$ for, both, periodic and non-periodic inputs. In practice, we split the input vector to the transformation into periodic and non-periodic inputs and apply $g_\theta$ and $\tilde{g}_\theta$ separately to each part. This also implies that we have to keep track of the permutations between coupling blocks, to be able to determine the type (periodic or non-periodic) of each input throughout the network. As a last detail, we use a uniform latent space distribution instead of a Gaussian for periodic dimensions.

Bayesian neural networks allow us to efficiently control the training stability and estimate training-related uncertainties. They extend standard architectures for regression, classification, or generative networks to distributions for each network weight, providing a key tool for explainable AI for instance in fundamental physics applications. The uncertainty on the output can then be extracted through sampling [67, 74–77]. For generative networks, the uncertainty can be defined for the underlying density estimation [3, 50, 68, 69], for which the network learns an uncertainty map over the target phase space. The critical aspect of Bayesian networks is how to make them numerically viable and still retain all their promising features. We use a variational approximation for the training, combined with independent Gaussians for each network weight. Such Bayesian networks include an optimal regularization, so they can outperform their deterministic counterparts with limited extra numerical effort. As always, we emphasize that the underlying approximations do not have to limit the expressivity of the networks when it comes to the sampled uncertainties. Moreover, we can treat the formal bias in the Gaussian widths as a hyperparameter, which needs to be adjusted and should be checked for stability.

We use the standard Bayesian version of the cINN, as introduced in Ref. [69], but with periodic splines. The network is implemented in PYTORCH [78]. In addition, we use the ADAM [79] optimizer with a constant learning rate. The hyper-parameters employed in our study are provided in Tab. 1.

## 2.3   Phase space parametrization

The unfolding method introduced above is identical to full, high-dimensional unfolding to the parton-level. However, in this application, we will only target a small number of kinematic distributions. Moreover, the unfolding network will then be used to define these distributions as part of the standard LHC analysis chain. This application allows us to improve the description of relevant phase space directions at the potential expense of correlations which are not useful for the measurement. In our case, we will guarantee the correct descriptions of the intermediate top-mass peaks through the network architecture.

The simplest way of encoding LHC events at the parton-level is through the components of the final-state 4-momenta. However, the corresponding redundant degrees of freedom are not adapted to the production of intermediate on-shell particles and it's reduced phase space. One way to improve the performance of generative networks is to add a maximum mean discrepancy (MMD) between a given set of generated and truth distributions [80] in the loss function. Its main advantage is that it only affects the target distribution and avoids an unnecessarily large model dependence. The disadvantage is that the additional loss term complicates the training and consequently limits the precision of the network. For our INN architecture, the computation of an MMD loss requires samples generated from the latent distribution, while the usual INN loss works on latent-space samples.

In our case, where the dominant signal and background processes share intermediate mass peaks, we can learn these features directly, through an appropriate phase space parametrization. For top decays with 9 degrees of freedom in the final state, a natural parametrization starts with the corresponding top 4-momentum, and then adds the invariant $W$-mass and a set of less sensitive angular observables,

$$\left\{ m_t, p_{T,t}, \eta_t, \phi_t, m_W, \eta_W^t, \phi_W^t, \eta_{\ell,u}^W, \phi_{\ell,u}^W \right\} . \tag{6}$$

Here $m_{t(W)}$ indicates the reconstructed invariant mass of the corresponding resonance. The superscripts $t$ and $W$ indicate the rest frame where the observable is defined, otherwise we use the laboratory frame. The indices $\ell$ and $u$ indicate the charged lepton and the up-type quark for leptonic or hadronic $W$-decays.

A network trained on this parametrization will reproduce the invariant top and $W$-mass distributions, but with drawbacks in the correlations of the hadronic $W$-decay. To extract $CP$-information, we also want to give the network access to the most important $CP$-observables, which we will discuss in detail in Sec. 3.1. This means we will include the Collins-Soper angle $\theta_{\text{CS}}$ [23, 28, 29, 44] and the angle between the charged lepton and the down-quark $\Delta\phi_{\ell d}$. One such parametrization for the entire $t\bar{t}$ system with 18 degrees of freedom is

$$\begin{aligned} \Big\{ \; &\vec{p}_{t\bar{t}}, m_{t_\ell}, |\vec{p}_{t_\ell}^{\text{CS}}|, \theta_{t_\ell}^{\text{CS}}, \phi_{t_\ell}^{\text{CS}}, m_{t_h}, \\ &\text{sign}(\Delta\phi_{\ell\nu}^{t\bar{t}}) m_{W_\ell}, |\vec{p}_\ell^{t\bar{t}}|, \theta_\ell^{t\bar{t}}, \phi_\ell^{t\bar{t}}, |\vec{p}_\nu^{t\bar{t}}|, \\ &\text{sign}(\Delta\phi_{du}^{t\bar{t}}) m_{W_h}, |\vec{p}_d^{t\bar{t}}|, \theta_d^{t\bar{t}}, \Delta\phi_{\ell d}^{t\bar{t}}, |\vec{p}_u^{t\bar{t}}| \; \Big\} . \end{aligned} \tag{7}$$

The superscripts CS and $t\bar{t}$ indicate the Collins-Soper frame of the $t\bar{t}$-system and the $t\bar{t}$ rest frame; the latter rotated such that $\vec{p}_{t_\ell}^{t\bar{t}}$ points in the direction of the positive $z$-axis. Also, $t_\ell$ and $t_h$ denote the leptonically and hadronically decaying tops, while $u$ and $d$ denote the up- and down-quarks from the $W$-decay. Using $\text{sign}(\Delta\phi_{AB}^{t\bar{t}})m_W$ as a phase space direction makes it harder for the network to generate the $W$-peaks, but solves the problem of quadratic phase space constraints.

We emphasize that the combination of generative unfolding with the phase space parametrization of Eq. (7) is expected to introduce a bias in the unfolding. However, for our application, we can ignore this bias given our choice of signal channel and our choice of target observable. Moreover, a potential bias will render the network-defined observable sub-optimal, but does not affect its evaluation in a standard analysis.

## 3 CP-phase from Higgs-top production

The example we choose to illustrate unfolding as a way to define dedicated observables is associated Higgs and top quark pair production

$$pp \to t\bar{t}h + \text{jets} \to (bu\bar{d})\,(\bar{b}\ell^-\bar{\nu})\,(\gamma\gamma) + \text{jets}\,, \tag{8}$$

plus the charge-conjugated process. $CP$-violating BSM effects modifying the top Yukawa coupling can be parametrized through the Lagrangian [81]

$$\mathscr{L} \supset -\frac{m_t}{v}\kappa_t\,\bar{t}(\cos\alpha + i\gamma_5\sin\alpha)th\,, \tag{9}$$

where $\alpha$ is the $CP$-violating phase, $\kappa_t$ the absolute value of the top Yukawa coupling, and $v = 246$ GeV the Higgs VEV. The SM-limit is $\kappa_t = 1$ and $\alpha = 0$. Deviations from the SM will affect Higgs production and the decay. While changes in the scalar Higgs decay will only impact the total rate, we focus on kinematic effects in the production.

The Lagrangian in Eq. (9) can be linked to the standard SMEFT framework used for general LHC analyses at mass dimension six. In this case, we introduce two Wilson coefficients to modify the top Yukawa [82, 83]

$$
\begin{aligned}
\mathscr{L} &\supset \frac{f_t}{\Lambda^2}\mathcal{O}_t + \frac{\tilde{f}_t}{\Lambda^2}\tilde{\mathcal{O}}_t \\
&\equiv \left(\phi^\dagger\phi - \frac{v^2}{2}\right)\left(\frac{f_t}{\Lambda^2}\left(\bar{q}_L t_R\tilde{\phi} + \tilde{\phi}^\dagger\bar{t}_R q_L\right) + i\frac{\tilde{f}_t}{\Lambda^2}\left(\bar{q}_L t_R\tilde{\phi} - \tilde{\phi}^\dagger\bar{t}_R q_L\right)\right),
\end{aligned} \tag{10}
$$

where $\phi$ is the Higgs doublet, $\tilde{\phi} = i\sigma_2\phi^*$, and $q_L$ the heavy quark doublet $(t_L, b_L)$. The parameters $\kappa_t$ and $\alpha$ in Eq. (9) can be computed as

$$\kappa_t^2 = \left(-1 + \frac{v^3 f_t}{\sqrt{2}m_t\Lambda^2}\right)^2 + \left(\frac{v^3\tilde{f}_t}{\sqrt{2}m_t\Lambda^2}\right)^2 \qquad \text{and} \qquad \tan\alpha = \frac{\tilde{f}_t}{f_t - \frac{\sqrt{2}m_t\Lambda^2}{v^3}}\,. \tag{11}$$

We emphasize that the SMEFT description implicitly assumes that new physics enters through higher-dimensional operators. In contrast, a $CP$-phase of the top Yukawa can already arise as a dimension-4 modification of the SM-Lagrangian, reflected by the scale combination $v^3/(m_t\Lambda^2)$ appearing above.

### 3.1 CP-observables

There are, fundamentally, two ways of testing the $CP$-structure of the Higgs Yukawa coupling introduced in Eq. (9): we can measure the angle $\alpha$ and conclude from a significant deviation $\alpha \neq 0$ that $CP$ is violated in the top Yukawa coupling, ideally using simulation-based inference [29, 30, 46, 84] or the matrix element method [50]. Alternatively, we can define an optimal observable for $CP$-violation and test the actual symmetry [12, 23, 42].

**Classical reconstruction**

To search for $CP$-violation, spin correlations between the top and anti-top quarks in $t\bar{t}h$ production are ideal, because the short top-lifetime allows for a transfer of the top-polarization to the decay products prior to hadronization or spin decorrelation [85]. The angular correlation between the top-spin and the momenta of the top decay products is given by

$$\frac{1}{\Gamma_t}\frac{d\Gamma}{d\cos\xi_i} = \frac{1}{2}\left(1 + \beta_i P_t \cos\xi_i\right),\tag{12}$$

where $\xi_i$ is the angle between the top spin and the $i$-th particle in the top quark rest frame, $P_t \in [0,1]$ is the polarization of the top quark, and $\beta_i$ is the spin analyzing power of the $i$-th decay product. Due to the left-handed nature of the weak interaction, the charged lepton and $d$-quark display the largest spin analyzing power,

$$\beta_{\ell^+} = \beta_{\bar{d}} = 1 \qquad \text{(to leading order)}.\tag{13}$$

While one cannot tag a $d$-jet, it is possible to find efficient proxies. A practical solution is to select the softer of the two light-flavor jets in the top rest frame. This choice gives a spin analyzing power for this jet as 50% of that of the charged lepton [29, 86, 87]. Assuming that the softer $W$-decay jet in the top rest frame comes from the $d$-quark, we can now construct appropriate angular correlations to measure.

**Linear CP-observables**

The basis of optimal observables testing a symmetry are $U$-even or $U$-odd observables, defined through their transformation properties on the incoming and outgoing states,

$$\mathcal{O}\left(U\left|i\right\rangle \to U\left|f\right\rangle\right) = \pm\mathcal{O}\left(\left|i\right\rangle \to \left|f\right\rangle\right),\tag{14}$$

where in our case $U = CP$. Furthermore, a genuine $U$-odd observable is defined as an observable which vanishes in a $U$-symmetric theory

$$\left\langle\mathcal{O}\right\rangle_{\mathscr{L}=U\mathscr{L}U^{-1}} = 0.\tag{15}$$

The two definitions are related in that any $U$-odd observable is also a genuine $U$-odd observable under the condition that the initial state and the phase space are $U$-symmetric [12, 88], so the genuine $U$-odd property is weaker.

Unfortunately, we cannot infer a $CP$-invariant theory from $\left\langle\mathcal{O}\right\rangle$ of a $CP$-odd observable alone. While a $\left\langle\mathcal{O}\right\rangle \neq 0$ always points to a $CP$-violating theory, the result $\left\langle\mathcal{O}\right\rangle = 0$ can appear in $CP$-symmetric and in $CP$-violating theories. To further analyze this case, we can construct a $CP$-odd observable that is also odd under the so-called naive time reversal $\hat{T}$. Now, the expectation value of this observable is completely tied to the $CP$-symmetry of the underlying theory [12].

$CP$-odd observables can be constructed either as $\hat{T}$-even scalar products of two 4-momenta or as a $\hat{T}$-odd contraction of four independent 4-momenta through the Levi-Civita tensor. For the $t\bar{t}$-system of $t\bar{t}h$ production we can use two top momenta and decay momenta

$$\left\{p_{b_l}, p_l, p_\nu, p_{b_h}, p_u, p_d\right\}.\tag{16}$$

It is straightforward to construct the $C$-even, $P$-odd, and $\hat{T}$-odd observable

$$\mathcal{O} = \varepsilon_{\mu\nu\sigma\rho}p_{t_h}^\mu p_{t_\ell}^\nu p_A^\rho p_B^\sigma,\tag{17}$$

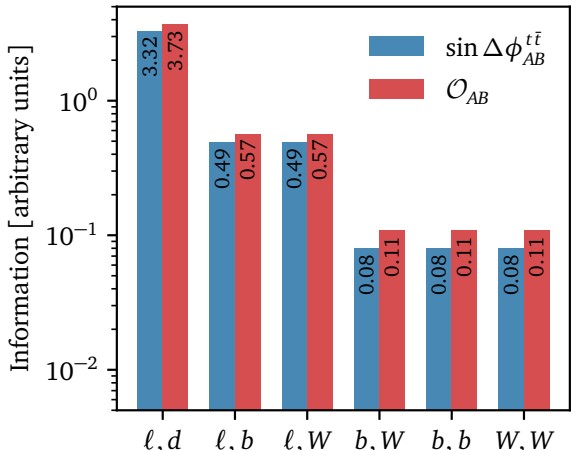

Figure 2: Fisher information $I$ for the linear CP-observables $\sin \Delta\phi_{AB}^{t\bar{t}}$ (blue) and $\mathcal{O}_{AB}$ (red), probing the sensitivity to $CP$-violating phase $\alpha$ in $t\bar{t}h$ production.

with suitable top decay momenta $p_{A,B}$. We can use the $CP$-invariance of the initial state and the phase space for $t\bar{t}h$ production to show that its expectation value in Eq. (17) vanishes in the SM.

In the $t\bar{t}$ center of mass frame, we can turn Eq. (17) into a triple product, a standard form for $CP$-odd observables,

$$\mathcal{O} = 2E_{t_h}\,\vec{p}_{t_\ell} \cdot (\vec{p}_A \times \vec{p}_B)\,. \tag{18}$$

However, it depends on the top 4-momentum, which are hard to determine accurately. It can be modified by introducing the azimuthal angle difference $\Delta\phi_{AB}^{t\bar{t}} = \phi_A^{t\bar{t}} - \phi_B^{t\bar{t}}$ in the $t\bar{t}$ frame [23, 29],

$$\Delta\phi_{AB}^{t\bar{t}} = \text{sgn}[\vec{p}_{t_\ell} \cdot (\vec{p}_A \times \vec{p}_B)]\arccos\left[\frac{\vec{p}_{t_\ell} \times \vec{p}_A}{|\vec{p}_{t_\ell} \times \vec{p}_A|} \cdot \frac{\vec{p}_{t_\ell} \times \vec{p}_B}{|\vec{p}_{t_\ell} \times \vec{p}_B|}\right], \tag{19}$$

to give us

$$\mathcal{O} = 2p_t^z E_t\, p_{T,A}\, p_{T,B}\, \sin \Delta\phi_{AB}^{t\bar{t}}\,, \tag{20}$$

where we choose $p_{t_\ell} = \{E_t, 0, 0, p_t^z\}$ and $p_{t_h} = \{E_t, 0, 0, -p_t^z\}$. By construction, $\mathcal{O}$ and $\Delta\phi_{AB}^{t\bar{t}}$ are sensitive to the linear interference terms in the scattering cross section, and therefore sensitive to the sign of the $CP$-phase.

These linear $CP$-observables can be constructed for various $\{A, B\}$ pairs, and their $CP$-sensitivity dependents on the spin-analyzing power of the particles $A$ and $B$. We compute the Fisher information metric $I$ to rank their $CP$-sensitivity, using MadMiner [29, 46]. The $\alpha$-dependent component of $I$ is defined as

$$I = \mathbb{E}\left[\frac{\partial \log p(x|\kappa_t, \alpha)}{\partial \alpha} \frac{\partial \log p(x|\kappa_t, \alpha)}{\partial \alpha}\right], \tag{21}$$

where $p(x|\kappa_t, \alpha)$ represents the likelihood of a phase space configuration $x$ given the theory parameters $\kappa_t$ and $\alpha$. $\mathbb{E}$ denotes the expectation value at the SM point, $(\kappa_t, \alpha)_{\text{SM}} = (1, 0)$. In Fig. 2, we show the Fisher information at parton-level associated with the linear $CP$-observables $\mathcal{O}_{AB}$ in red and the Fisher information for $\sin \Delta\phi_{AB}^{t\bar{t}}$ in blue.

First, we see that for all combinations $(A, B)$ the Fisher information in $\mathcal{O}_{AB}$ is slightly larger than the Fisher information in $\sin \Delta \phi_{AB}^{t\bar{t}}$, an effect of the momentum-dependent prefactor in $\mathcal{O}_{AB}$. Among the various combinations $(A, B)$, the combination of the lepton and the down-type quark is the most sensitive. This corresponds to the maximal spin analyzing power for this pair. Next comes the combination where either the charged lepton or the down quark is replaced by the $b$-quark or the $W$-boson. In this case, the Fisher information is suppressed by two powers of

$$\beta_b = \beta_W \sim 0.4 \, . \tag{22}$$

The correlation between a pair of $b$-quarks or $W$-bosons is further suppressed by another factor $\beta_{b,W}^2$.

**Non-linear observables and Collins-Soper angle**

For a given realization of $CP$-violation in an SM-like interaction vertex, the $CP$-observable defined in the previous section is not guaranteed to be the most powerful observable [10]. This is obvious for dimension-6 operators, where the symmetry structure is often combined with a momentum dependence of the interaction [12], and the two aspects can, in principle, be tested independently. Comparing the two handles, $CP$-odd observables are only sensitive to the interference between the SM-contribution and the $CP$-violating matrix element, while observables testing the momentum structure of the interaction vertex can be dominated by the new-physics-squared contribution. For large $CP$-phases $\alpha$, the more promising analysis strategy will use a general test of the structure of the top-Higgs coupling. This motivates using a combination of dedicated $CP$-observables with general interaction probes as an optimal search strategy.

Several observables have been evaluated as probes of the $CP$-phase $\alpha$ in Eq. (9) using $t\bar{t}h$ production [23, 28, 29]. They include the pseudorapidity difference between the two tops and the azimuthal angle between the two tops, the Higgs transverse momentum [81, 89], or the invariant mass of the top and anti-top pair,

$$\left\{ \Delta \eta_{t\bar{t}}, \Delta \phi_{t\bar{t}}, p_{T,h}, m_{t\bar{t}} \right\} \, . \tag{23}$$

These standard observables can be supplemented with the projection angle [81, 89, 90]

$$b_4 = \frac{p_{z,t}}{|\vec{p}_t|} \cdot \frac{p_{z,\bar{t}}}{|\vec{p}_{\bar{t}}|} \, . \tag{24}$$

Finally, we can use the Collins-Soper angle $\theta_{\text{CS}}$ [44], the angle between the top quark and the bisector of the incoming hadrons in the $t\bar{t}$ center of mass frame. The original motivation for the Collins-Soper angle was to define an observable for the Drell-Yan process $pp \to \ell^+ \ell^-$ that corresponds to the scattering angle. Factorization arguments suggest the di-lepton rest frame, to minimize ISR-effects and then study the angular correlations between the incoming quarks and the outgoing leptons. In this frame the 3-momenta of the quarks and leptons each define a plane, and in turn an azimuthal angle and a polar angle between the two planes.

Without ISR the $z$-axis of the so-defined CS-frame is trivially given by the parton and hadron directions. Including ISR, we instead define this $z$-axis as halving the angle between one of the hadrons and the reverse direction of the other hadron. The Collins-Soper angle can be used to measure the polarization of the intermediate gauge boson, the weak mixing angle [91], or the (Lorentz) structure of the interaction vertices. The Collins-Soper angle can also be used to probe the structure of the Higgs-photon coupling [92, 93] and to boost new

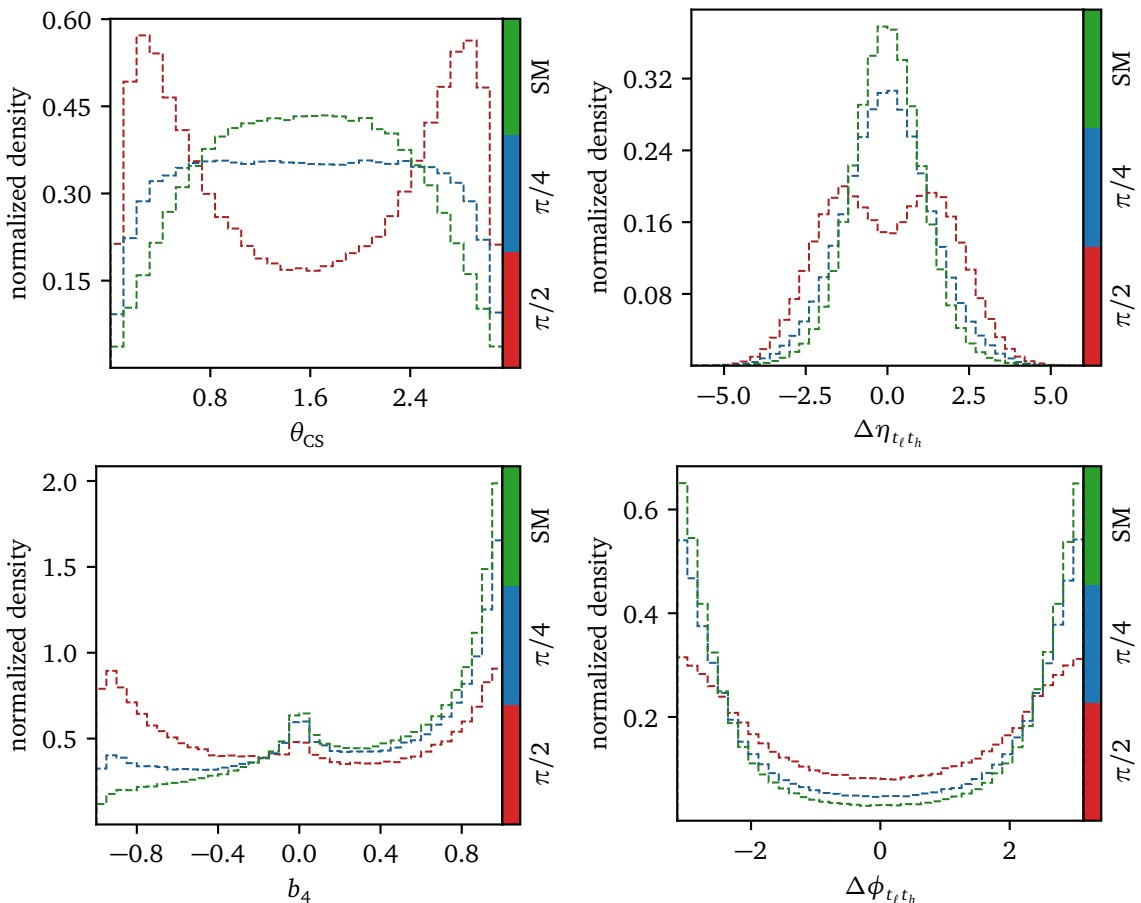

Figure 3: Distributions for the CS-angle $\theta_{\rm CS}$, $\Delta\eta_{t_\ell t_h}$, $b_4$, and $\Delta\phi_{t_\ell t_h}$, all at parton-level for semileptonic $t\bar{t}h$ production. The distributions are shown for SM (green), $\alpha = \pi/4$ (blue) and $\alpha = \pi/2$ (red).

physics searches in $h^* \to ZZ$, $Zh$, and $t\bar{t}Z$ channels [94–97]. Finally, it can be generalized to $t\bar{t}$ production, where it is constructed for the top momentum in the $t\bar{t}$ rest frame [23,98,99]. While the Collins-Soper angle has no specific sensitivity to $CP$-violation, we view it as the Swiss Army knife of coupling tests.

All above-mentioned kinematic observables are sensitive to the new-physics-squared terms, proportional to $\sin^2\alpha$ or $\cos^2\alpha$, in the $t\bar{t}h$ rate, with no sensitivity to the sign of the CP-phase. From Ref. [29], we know the relative sensitivity of these observables to probe the Higgs-top $CP$-structure through a modified Fisher information metric, accounting for non-linear effects. The top-five observables with the highest Fisher information for $\alpha$ are (symbolically written)

$$\Delta\eta_{t\bar{t}} > \theta_{\rm CS} > b_4 > \Delta\phi_{t\bar{t}} > p_{T,h}\,. \tag{25}$$

We show the parton-level distributions for the four most sensitive observables in the semileptonic $t\bar{t}h$ channel for the SM value $\alpha = 0$ and $\alpha = \pi/4, \pi/2$ at the LHC with $\sqrt{s} = 14$ TeV in Fig. 3. Different values of $\alpha$ lead to distinctly different profiles in the distributions.

As alluded to above, the technical challenge and a limitation to the optimality of a given analysis is to construct the different observables in their respective kinematic frames. Considering their strong sensitivity on $\alpha$, we include the leading observables in the phase space parametrization given in Eq. (7) to target this problem directly.

## 3.2 Unfolding-based analysis

The standard challenge for every LHC analysis is to extract a small signal from a large (continuum) background. For our simple study, we show how we can avoid modeling this step. The generative unfolding trained on $t\bar{t}h$ events gives us the probability $p(x_{\text{part}}|x_{\text{reco}}, S)$ that a parton-level signal event $x_{\text{part}}$, corresponds to an assumed signal event $x_{\text{reco}}$ at reconstruction level. What we are ultimately interested in, however, is a model parameter $\alpha$, which could be a mass, a $CP$-phase, or any other continuous theory parameter, which affects our signal distribution. Since we do not know if a particular reco-level event $x_{\text{reco}}$ is signal or background, we only care about the full probability $p(\alpha|x_{\text{reco}})$ of our model parameter, given some reco-level event $x_{\text{reco}}$ which is either signal or background. Since $\alpha$ does not change the background, this probability can be split into the distribution $p(\alpha|x_{\text{part}})$, where $x_{\text{part}}$ is a parton-level signal event, and the probability $p(x_{\text{part}}|x_{\text{reco}})$ of $x_{\text{part}}$ given $x_{\text{reco}}$:

$$p(\alpha|x_{\text{reco}}) = \int p(\alpha|x_{\text{part}})p(x_{\text{part}}|x_{\text{reco}}) \, \mathrm{d}x. \tag{26}$$

The challenge is to compute $p(x_{\text{part}}|x_{\text{reco}})$ from our unfolding result $p(x_{\text{part}}|x_{\text{reco}}, S)$. Using the definition of conditional probabilities we can write

$$\begin{aligned} p(x_{\text{part}}|x_{\text{reco}}) &= \sum_{T \in \{S, B\}} p(x_{\text{part}}|x_{\text{reco}}, T)p(T|x_{\text{reco}}) \\ &= p(x_{\text{part}}|x_{\text{reco}}, S)p(S|x_{\text{reco}}) + p(x_{\text{part}}|x_{\text{reco}}, B)(1 - p(S|x_{\text{reco}})), \end{aligned} \tag{27}$$

where the probabilities of $x_{\text{reco}}$ being a signal or background event, $p(T|x_{\text{reco}})$, can be encoded in a trained classifier. Let us consider for a moment what the probability $p(x_{\text{part}}|x_{\text{reco}}, B)$ tells us. We are interested in signal events $x_{\text{part}}$, i.e. events that are affected by $\alpha$. By definition, background events $x_{\text{reco}}$ cannot give us any information, beyond prior knowledge, about $x_{\text{part}}$. For this reason, we can drop $x_{\text{reco}}$ and write $p(x_{\text{part}}|x_{\text{reco}}, B) = p(x_{\text{part}})$, where $p(x_{\text{part}})$ is only constrained through prior knowledge. This includes our model assumptions as well as phase-space constraints due to a finite center-of-mass energy. We can now write

$$p(x_{\text{part}}|x_{\text{reco}}) = p(x_{\text{part}}|x_{\text{reco}}, S)p(S|x_{\text{reco}}) + p(x_{\text{part}})(1 - p(S|x_{\text{reco}})). \tag{28}$$

What Eq. (28) shows, is that we can limit our unfolding model to extracting $p(x_{\text{part}}|x_{\text{reco}}, S)$ and still include background events into our analysis later, without changing our model.

As given in Eq. (8), we study $pp \to t_h \bar{t}_\ell h$ production with $h \to \gamma\gamma$ at the HL-LHC. The dominant background is continuum $t\bar{t}\gamma\gamma$ production, subdominant contributions arise from the process $Wb\bar{b}(h \to \gamma\gamma)$. We use MadGraph5_aMC@NLO [100] with NNPDF2.3QED [101] to generate signal events at leading order with $\sqrt{s} = 14$ TeV. Signal events are simulated without kinematic cuts using the HC_NLO_X0 UFO model [102, 103]. Parton showering and hadronization effects are simulated using Pythia 8 [104]. The detector response is simulated with Delphes 3 [105], using the default ATLAS HL-LHC card [106, 107].

Next, we select events containing exactly two photons, two $b$-tagged jets, one lepton, and at least two light-flavored jets. The individual particles in the final state are required to satisfy the acceptance cuts

$$\begin{aligned} p_{T,b} &> 25 \text{ GeV}, & p_{T,j} &> 25 \text{ GeV}, & p_{T,\ell} &> 15 \text{ GeV}, & p_{T,\gamma} &> 15 \text{ GeV}, \\ |\eta_b| &< 4, & |\eta_j| &< 5, & |\eta_\ell| &< 4, & |\eta_\gamma| &< 4. \end{aligned} \tag{29}$$

At the parton-level, the signal phase space involves eight final state particles; following Sec. 2.3 it requires 22 parameters if we are assuming the Higgs is fully and uniquely reconstructed.

The training dataset involves an event-wise pairing of parton and detector level events with up to six light-flavored jets, satisfying the selection cuts in Eq. (29). While the event at the reconstruction level requires additional degrees of freedom for jet radiation, the number of degrees of freedom is reduced by the neutrino. An additional challenge is the combinatorics of the $b$-jets and light-flavor jets.

### 3.3 Results

**Jet combinatorics** The first results from unfolding $t\bar{t}h$ SM-events are presented in Fig. 4. We train the unfolding network on SM-events and also apply it to SM-events. First, we examine the robustness of the network to unfold a variable number of jets to the parton-level. For our lepton-hadron reference process in Eq. (8) two light-flavor jets come from the hadronic top decay, while additional jets arise from QCD jet radiation. The unfolding network has to reconstruct the two hard jets at the parton level from a variable number of jets at the detector level [60].

To evaluate the unfolding performance, we examine four invariant masses: $m_{t_\ell}$, $m_{t_h}$, $m_{ht_h}$, and $m_{t_\ell t_h}$. We train one network on SM events without ISR and one network on events with up to six light-flavored jets. The corresponding cINN-generated distributions are shown as

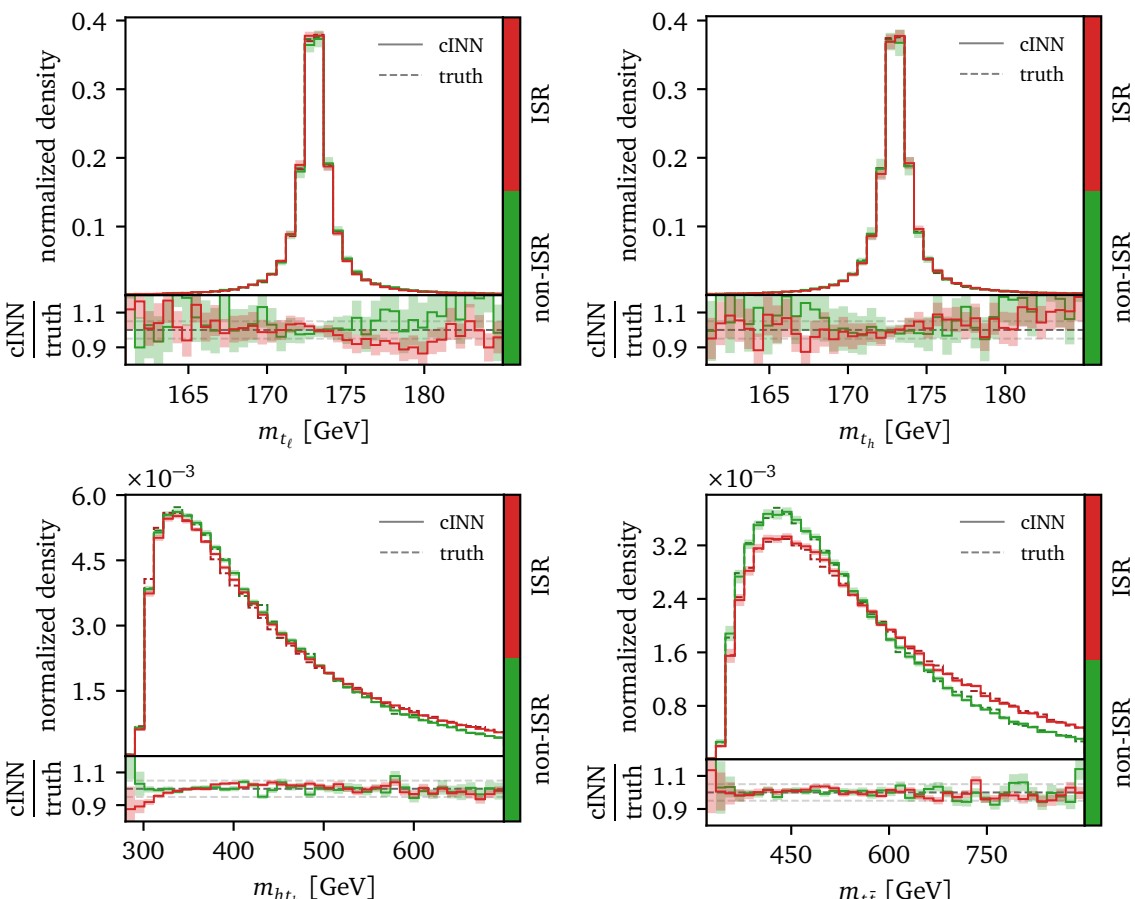

Figure 4: Jet combinatorics — cINN-generated distributions for $m_{t_\ell}$, $m_{t_h}$, $m_{ht_h}$ and $m_{t_\ell t_h}$ in the SM. Unfolded distributions are shown as solid lines, parton-level truth as dashed lines. The training data set either does not include ISR (green) or up to six ISR jets (red).

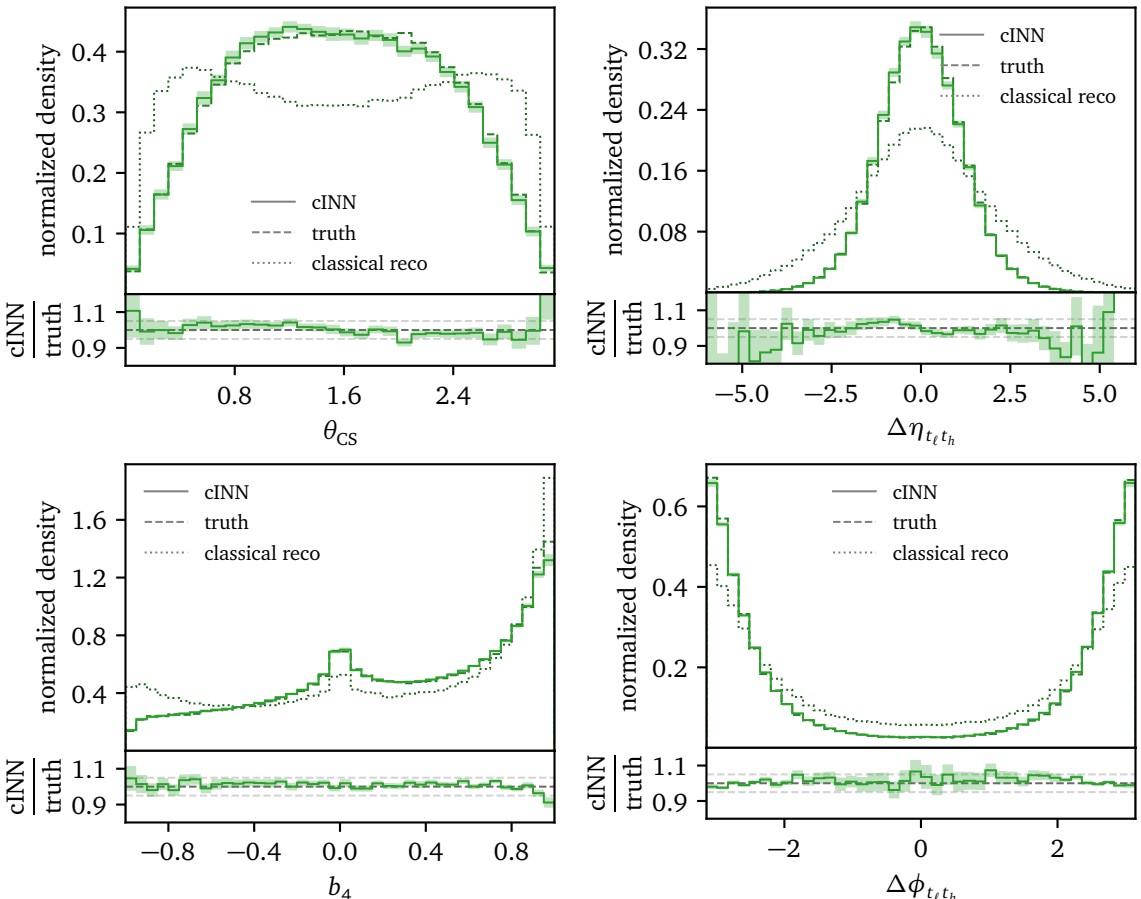

Figure 5: Reconstructing dedicated observables — cINN-generated distributions and distributions based on classical reconstruction [29] for $\theta_{CS}$, $\Delta\eta_{t_\ell t_h}$, $b_4$ and $\Delta\phi_{t_\ell t_h}$ for SM events. The secondary panels show the bin-wise agreement between the cINN-generated distributions and the parton-level truth.

solid lines in Fig. 4. The parton-level truth is displayed as dashed lines. We find that unfolded distributions generated by both networks are in good agreement with the parton-level truth in the bulk of the phase space. Despite the added combinatorial ambiguity, the performance of both networks is largely comparable. We also show the uncertainties from the Bayesian setup, represented as $1\sigma$ error bands. They test the stability of the unfolding network similar to an ensemble of networks. It is important to observe that the truth distributions remain within these error bands.

**Reconstructing dedicated observables**  For Fig. 5 we train the unfolding network on SM events with up to six light-flavor jets. We compare cINN-generated events at the parton-level and in the appropriate rest frame with events from a classical reconstruction for four particularly interesting observables from Sec. 3.1: $\theta_{CS}$, $\Delta\eta_{t\bar{t}}$, $b_4$, and $\Delta\phi_{t\bar{t}}$. For comparison, we display the parton-level truth as dashed lines. In the ratio we observe that the generated distributions agree with the truth within a few percent. Slightly larger deviations in the tails are due to limited training statistics.

The conventional approach to complex kinematic correlations in the semileptonic $t\bar{t}$ system relies on a complex reconstruction algorithm, with a significant loss of information due to missing correlations [29]. We show the reconstructed distributions from the classical recon-

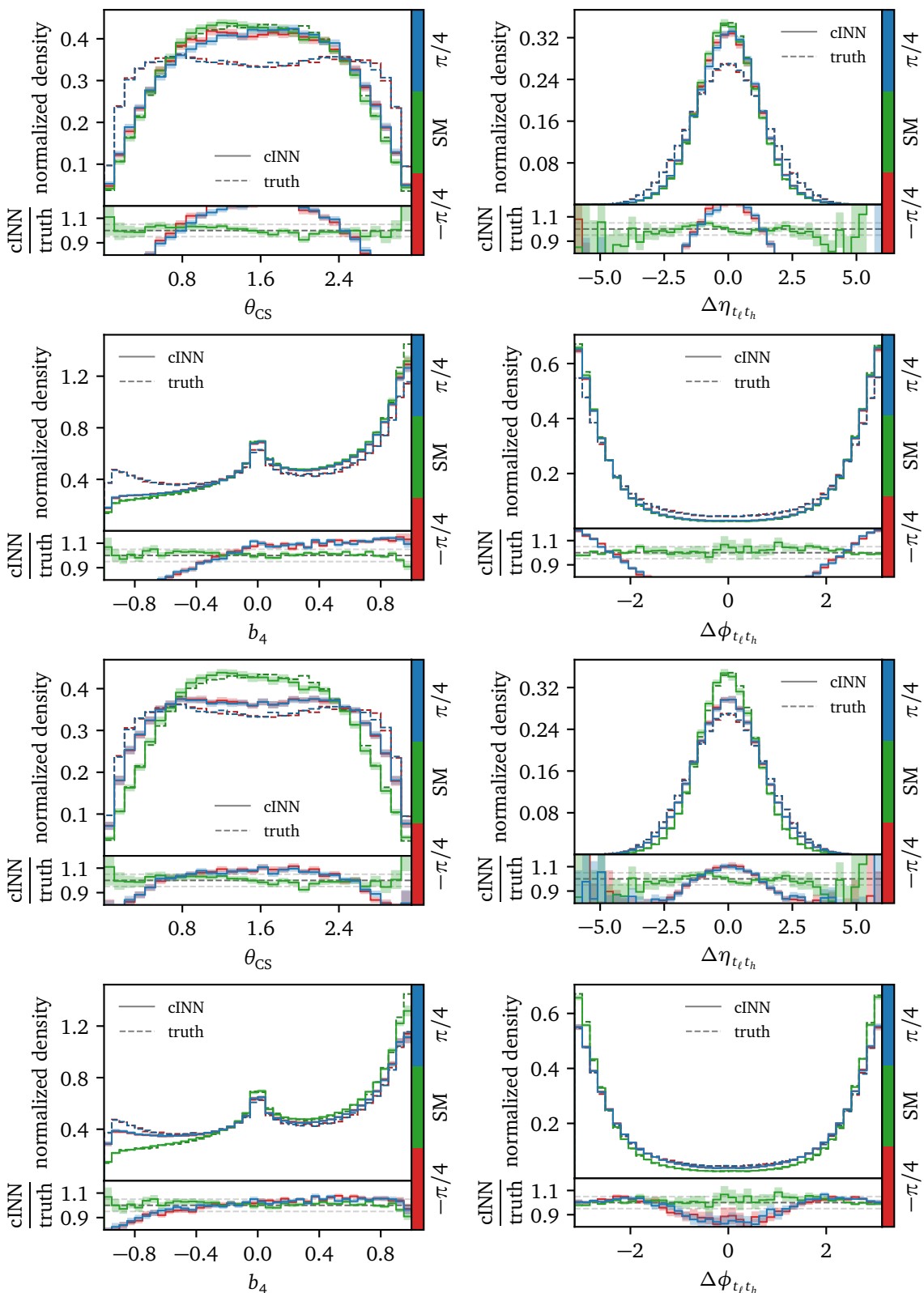

Figure 6: Model dependence — cINN-generated distributions for $\theta_{CS}$, $\Delta\eta_{t_\ell t_h}$, $b_4$, and $\Delta\phi_{t_\ell t_h}$. Upper two rows: unfolding of SM events using three different networks, trained on data with $\alpha = -\pi/4, 0, \pi/4$. Lower two rows: unfolding of events with $\alpha = -\pi/4, 0, \pi/4$, with a network trained on SM events.

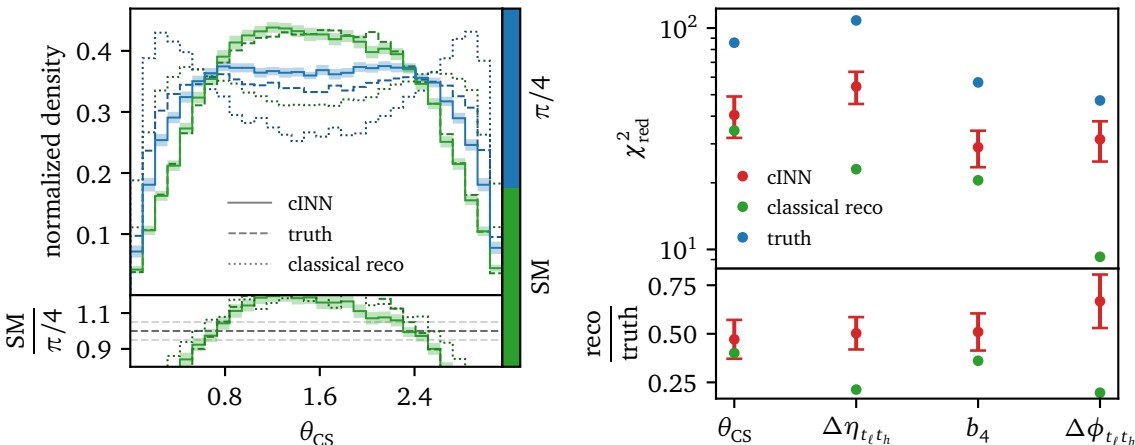

Figure 7: Sensitivity — Left: cINN-generated distributions for $\theta_{CS}$ from unfolding events with two $\alpha$ values. These generated distributions are compared to the distributions obtained from classical reconstruction methods, as described in Ref. [29], and the respective truth. Right: To quantify the sensitivity of the cINN, as shown here for $\theta_{CS}$, we compute the reduced $\chi^2$-value between the distributions ($\sim$120k events and 64 bins) for both $\alpha$ values, using the Poisson errors of the bin counts. We do this for the cINN-generated (red), classically reconstructed (green), and truth distributions (blue) of $\theta_{CS}$, $\Delta\eta_{t_\ell t_h}$, $b_4$, and $\Delta\phi_{t_\ell t_h}$. In the bottom panel, we also show the ratio of the reduced $\chi^2$-values of the cINN and the classical reconstruction to the truth. Uncertainties on cINN-generated values are obtained from the Bayesian setup.

struction strategy developed in Ref. [29] as dotted lines. Comparing these distributions to the cINN-unfolded version, we see that at least for a network trained and tested on SM events, the improvement from generative unfolding is striking.

**Model dependence**     After observing the significant improvement through our new method for SM events, we need to test how model-dependent the network training is. In the upper panels of Fig. 6, we reconstruct the usual set of key observables for SM events, but with three different networks, trained on events generated with the *CP*-angles $\alpha = -\pi/4, 0, \pi/4$. We adopt the BSM values $\alpha = \pm\pi/4$ here, as these choices closely align with the current experimental limits [108, 109]. From Fig. 6 we expect, for instance, the network trained on events with $\alpha = \pi/4$ to be biased towards a broader $\theta_{CS}$ distribution, a wider rapidity difference $\Delta\eta_{t_\ell,t_h}$, and a flatter $b_4$ distribution. In the different panels we see a slight bias, especially in the secondary panels. But the bias remains at the order of 10%, at most 20%, much below the change in the corresponding distributions from varying $\alpha$. On the other hand, this bias is significantly larger than the uncertainty band, which indicates that this model dependence can be reduced through the proposed iterative method of Ref. [62]. The corresponding study is beyond the scope of this paper, because it balances a reduced bias of the unfolding with less statistics, an aspect which we do not include in this proof-of-principle study.

In the lower panels of Fig. 6 we test the model dependence the other way around, by unfolding data for different $\alpha = -\pi/4, 0, \pi/4$ using a network trained on SM events. The figure of merit are the ratios of cINN-unfolded and respective truth distributions, shown in the secondary panels. This situation is closer to the reality of a measurement, where we infer $\alpha$ by comparing the distribution extracted from data to different simulated hypotheses. As before, we see a slight bias, now towards the SM structure of a more narrow $\theta_{CS}$ distribution, a narrow rapidity difference $\Delta\eta_{t_\ell,t_h}$, and a steeper $b_4$ distribution. Also, as before, the effect

of the bias is much smaller than the effect of $\alpha$ on the data, leaving us optimistic that we can use the cINN-unfolded distribution to measure $\alpha$.

**Sensitivity**    Finally, in Fig. 7, we apply the generative unfolding to SM and $\alpha = \pi/4$ events. The unfolding network is trained on SM events. As a baseline comparison, we also show the same two curves for classical reconstruction of $\theta_{\text{CS}}$, following Ref. [29] as dotted lines in the left panel of Fig. 7. As mentioned earlier, generative unfolding leads to a major improvement over classical reconstruction. The difference in the two unfolded kinematic distributions, shown in solid lines, illustrates the reach of an analysis based on the kinematic distribution. To showcase the improvement in new physics sensitivity, we calculate the reduced $\chi^2$ values for $\theta_{\text{CS}}$, $\Delta\eta_{t_\ell t_h}$, $b_4$, and $\Delta\Phi_{t_\ell t_h}$ between the SM and $\alpha = \pi/4$ hypotheses, using the Poisson errors of the bin counts. The reduced $\chi^2$ values are computed with $\sim$120k events and 64 bins, for three scenarios: parton-level truth (blue), classical reconstruction from Ref. [29] (green), and the cINN-based generative model trained on SM events (red). A higher $\chi^2$ value indicates a greater sensitivity to new physics.

The results show that the unfolding setup leads to an enhancement in sensitivity compared to the classical reconstruction strategy. This indicates that the generative unfolding approach is effective in extracting more information from the kinematic distributions, thereby improving the analysis' capability to detect and explore new physics phenomena. We further observe that the network is slightly more consistent in reproducing the sensitivity relations of the true observable distributions than the classical reconstruction. The latter performs well on some observables, but quite bad for others. Especially surprising is the classical sensitivity on $\theta_{\text{CS}}$, given that the reconstruction here is far from the actual CS-angle.

# 4    Outlook

Unfolding is one of the most exciting development of analysis preservation and publication at the LHC. Modern machine learning makes it possible to unfold high-dimensional distributions, covering all correlations without binning. Generative unfolding defines this unfolding in a statistically consistent manner. However, using unfolded data is a challenge for the ATLAS and CMS analysis chains, especially in controlling and estimating uncertainties.

We investigated a simpler application of the unfolding technique, the extraction of a kinematic observable in a specific partonic reference frame. It solves the dilemma that on the one hand an optimal observable requires no complex correlations, but on the other hand such an observable is, typically, hard to reconstruct. In this case the generated kinematic distribution can be used like any other observable; the unfolding network is nothing but a kinematic reconstruction algorithm.

The perfect examples for a challenging kinematic correlation are the Collins-Soper angle or the optimal $CP$-observables in $t\bar{t}h$ production. They allow us to measure a $CP$-phase in the top Yukawa coupling, a cosmologically relevant parameter entering an LHC signature at dimension four and at leading order. We have shown that unfolding allows us to extract the leading observables for such a $CP$-phase $\alpha$, with the help of an appropriate phase space parametrization. While such a parametrization might shape the unfolded kinematic distribution, this effect can be controlled through calibration.

First, we have shown that the cINN-unfolding can solve the combinatorics of $W$-decay jets vs QCD jet radiation. Second, the unfolded distributions of SM events, with a network trained

on SM events, show excellent agreement with the parton-level truth. Potential differences are covered by the uncertainty estimate from the Bayesian network. Third, we have tested the model dependence in two different ways — unfolding SM event using networks trained on events with different amounts of $CP$-violation and unfolding events with $CP$-violation using a network trained on SM events. For the former, we have found that there exists a small, but significant model dependence, which can be removed through Bayesian iterative improvements. For the latter, the unfolded distributions do not perfectly reproduce the respective truth, but the bias is much smaller than the kinematic effect of the $CP$-angle.

All these tests have motivated a comparison of the reach of the HL-LHC for the $CP$-angle $\alpha$, based on classical reconstruction methods and on cINN-unfolded distributions. The generative unfolding approach effectively extracts more information from kinematic distributions, enhancing sensitivity to new physics phenomena. This highlights the importance of advanced machine learning techniques, such as cINNs, for the HL-LHC.

While this study is clearly not the last word on this analysis technique, we consider the outcome promising enough for an experimental study, with a proper treatment of statistical limitations, continuum backgrounds, calibration, and iterative improvements of the unfolding network.

## Acknowledgements

RKB and DG thank the U.S. Department of Energy for financial support, under grant number DE-SC0016013. Some computing for this project was performed at the High Performance Computing Center at Oklahoma State University, supported in part through the National Science Foundation grant OAC-1531128. TH is funded by the Carl-Zeiss-Stiftung through the project *Model-Based AI: Physical Models and Deep Learning for Imaging and Cancer Treatment*. This research is supported by the Deutsche Forschungsgemeinschaft (DFG, German Research Foundation) under grant 396021762 – TRR 257: *Particle Physics Phenomenology after the Higgs Discovery* and through Germany's Excellence Strategy EXC 2181/1 – 390900948 (the *Heidelberg STRUCTURES Excellence Cluster*).

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
