# Peer review of "Returning CP-Observables to The Frames They Belong"

_SciPost Physics_

## Round 1 · Referee Report · Anonymous (Referee 1) · 2023-10-4

Report

The authors discuss multi-dimensional unfolding of optimal kinematic observables for a measurement of a CP-phase in the top Yukawa coupling. Most of kinematic observables discussed in the paper are already introduced in literature but authors revisit them with cINN and showed the improved sensitivity with the unfolded distributions. This topic is certainly interesting and deserves attention. The article is well written and the analysis has been done carefully. Therefore I recommend publication of this article in SciPost.

I have a few minor (general) comments/suggestions that I would like the authors to consider. Paper is already in good shape and so I wouldn't request revision. I will leave up to authors whether or not they want to address some of these comments in their revised version but it would be beneficial for readers, if authors make some effort for some of comments below.

  1. Unfolding has been extensively used in HEP analyses and there are many (non NN-based) unfolding algorithms available, including iterative method, singular value decomposition, bin-by-bin correlation etc. Many of these methods already exist in a popular analysis code such as Root. Why does one needs cINN (or NN)? What would be advantages of using NN over existing unfolding methods? Is it clear that conventional unfolding methods might miss anything important that NN might capture? There are some comparison studies such as https://arxiv.org/pdf/2104.03036.pdf but it should be straightforward for authors to make a quick comparison for Top Yukawa CP phase.

  2. There are already several suggestions on using ML for unfolding. Especially the following references advertise multi-dimensional unfolding as authors study in their paper:

    https://arxiv.org/pdf/1911.09107.pdf https://arxiv.org/pdf/2203.16722.pdf

I understand that authors are focusing more on physics side (CP phase) and authors simply could have used conventional unfolding methods or existing ML-based unfolding. Are there particular reasons why cINN might be more suitable than these for CP phase study?

  1. I might have missed discussion but do authors perform unfolding with unbinned data or histograms? It seems that authors are doing unbanned analysis from the following sentence in conclusion "Modern machine learning makes it possible to unfold high-dimensional distributions, covering all correlations without binning." It may be good to make this clear in earlier sections, if not mentioned.

  2. I see PDF information for event generation but what scale is chosen?

  3. It seems that the (red) error bar in the right panel of Fig. 7 comes from statistical uncertainty. Wouldn't there be some systematic uncertainties coming from NN and from unfolding procedure? Is it obvious that such systematic uncertainties are much smaller than statistical uncertainty and therefore negligible?

  4. It is quite interesting to notice that classical reconstruction methods and cINN method are comparable for $\theta_{CS}$ and $b_4$. Do we know why?

  • validity: -
  • significance: -
  • originality: -
  • clarity: -
  • formatting: -
  • grammar: -

Author:  Rahool Kumar Barman  on 2024-03-23  [id 4375]

(in reply to Report 1 on 2023-10-04)
Category:
answer to question

We would like to thank the referee for carefully reading our manuscript and providing detailed and valuable comments. The version of the manuscript that we resubmit addresses the aspects that the report brought to our consideration. Please find the comments from the referee and our answers in the file attached below.
The file can also be accessed through this link: https://www.dropbox.com/scl/fi/6982jk2o0brtnaotk0oa8/Reply_to_report_1.pdf?rlkey=0l879piw1mlsfxx1ds9yoyivf&dl=0

Attachment:

Reply_to_report_1.pdf

---

## Round 1 · Referee Report · Florian Bury (Referee 2) · 2023-10-12

Strengths

1- Very promising results in terms the unfolding 2- Innovative method using ML 3- Easy to interpret figures and conveying the message effectively 4- Strong theoretical motivations for the choice of CP observables

Weaknesses

1- Lack of clear message and links between sections. The content of some sections are not justified enough for their presence in the paper (eg, the splines, the SMEFT section, or enumeration of variables not used later) 2- Lack of information prevents reproducibility of results from experts 3- The paper cannot be understood as standalone, and uses a lot of links to previous work

Report

This paper meets the requirement to be published in SciPost Physics. The method developed through cINN is cutting-edge, and although the proposed unfolding strategy is not new (as it has been presented previously by the authors), its application to CP-observables is original. The results seem sound, with a clear presentation, and extremely promising for future experimental searches. The paper however lacks details in the method implementation, some of them being available in previous publications, at a cost for the reader which has to follow through in the references. Certain sections do not seem to fit well, which may be just a matter of needing a more detailed justification, easily fixable from the author's side. Minor corrections are needed, mostly to help the reader's understanding, but the physics and logic are reasonable.

Requested changes

1- Section 2.1. The cINN is the salient feature of your method, yet it is barely introduced in this section. From reading the many references, it is possible to understand what you mean in eq (1) and (2), notaly the conditional part of the INN, but otherwise the paper cannot be understood alone. It is obvious that this paper builds up on your previous work, and the very short description of cINNs is made to shorten an already lengthy paper. However, for an average user that would not have read the references, this part is very much unclear and would benefit from a few more paragraphs (describing how the loss function is built from Bayes' theorem, and how the "s" and "t" networks are used in a coupling block for example). It does make a bit of redundancy for a reader that followed your previous work, but will improve the "standalone" readability of the paper 2- Section 2.2. The justification for the periodic spline is a bit lacking, and the link with sec 2.3 is absolutely not clear. Similarly, the choice of a uniform latent distribution (instead of the usual Gaussian) is not justified, unless it has a link with the periodicity, which emphasizes the need for more explanation. 3- Section 2.3. Could you shortly elaborate on the calculations that lead to 9 dof for a single top ? This section seems out of context compared to the rest of the paper. You list a set of variables that fully determine the system, yet afterwards only select a few for your unfolding (even some like b4 that were not in the list). Unless you apply a MMD lose to all these distributions? The MMD is also lacking some details (the reader can follow the reference of course, but it makes the reading again a bit uneasy). 4- Overall, the section 2 does not allow for easy reproducibility of the method, considering it is one of the acceptance criteria of the journal. This concerns the technical aspects of the cINN (see comments 1 and 2), but also of the training itself. 5- Section 3 intro (SMEFT). The rest of the paper being entirely about $\alpha$, this discussion about SMEFT seems unnecessary. Only the last sentence could justify you can have a CP-observable of dim-4 (thereby motivating the whole use of the method on ttH), but it is not really obvious from the phrasing. 6- Section 3.2. Could you elaborate on the fact that background events can be included later in the analysis ? Unless mistaken, all the plots are about signal events that are unfolded. How does this compare when the backgrounds are dominant, does it matter, and how does it affect the sensitivity ? 7- Section 3.3 (Jet combinatorics). You talk about variable number of jets, but nowhere in the text do you explain how you deal with this. Is it the same method as Ref [60] ? If so, this would be worth explaining a bit more, especially considering combinatorics are a major issue in most data analyses. 8 - Section 3.3 (Model dependence). Outside the iterative method, how could this bias be dealt with in an analysis? As a systematic uncertainty ? Does the fact that the cINN does not extrapolate to non-zero values of $\alpha$ when trained on SM mean that for each $\alpha$ that experimentalists want to try, a new cINN must be trained ? Would this not be a major shortcoming ? 9- Section 4. The assertion that the iterative method could remove the model bias is a bit strong. It is likely, but given that it has not been attempted here, better be a bit more conservative. 10- General comment. This method looks very promising and would be worth applying to either CMS or ATLAS analyses. Your paper is of course a proof of concept, so it does not address much the challenges met during a real life analysis (exp and th uncertainties, jet energy corrections, possibly datadriven methods, combinatorics, particles outside acceptance, etc), which you can of course not be blamed for. Still, when novel methods like this one are presented in a paper, it is always on a somewhat idealized scenario and for the case that works well (though granted, you consider complications like jet combinatorics with ISR). By experience, "no plan survives first contact with data", and there is always more complication when using novel methods in a real life data analysis. It would therefore be interesting to know your opinion or experience with the challenges such analyses could face that you did not have, when applied to data. It may probably fall outside your scope of the paper, you could therefore disregard this comment, but some considerations specifically targeted at experimentalists that would want to use your technique could potentially enrich your paper and make it more easily advertisable to that audience.

  • validity: high
  • significance: high
  • originality: good
  • clarity: good
  • formatting: excellent
  • grammar: perfect

Author:  Rahool Kumar Barman  on 2024-03-23  [id 4376]

(in reply to Report 2 by Florian Bury on 2023-10-12)
Category:
answer to question

We would like to thank the referee for carefully reading our manuscript and providing detailed and valuable comments. The aspects that the report brought to our consideration have been addressed in the version of the manuscript that we resubmitted. Please find the comments from the referee and our answers in the file attached below. The file can also be accessed through this link: https://www.dropbox.com/scl/fi/pcj9wd2ox539aax1bo3tp/Reply_to_report_2.pdf?rlkey=bi8ksng4e20pun4hbkjqbmeni&dl=0

Attachment:

Reply_to_report_2.pdf

---

## Round 1 · Referee Report · Prasanth Shyamsundar (Referee 3) · 2023-10-14

Strengths

1) A novel approach to use unfolding techniques for data analysis. 2) The analysis technique is well motivated and shows promising results.

Weaknesses

1) Several parts of the paper feel unclear, incomplete, and/or inaccurate. 2) The organization of the paper could be better.

Report

This work uses an ML-based unfolding technique to create collider analyses variables sensitive to CP-violation. This is a novel idea, and I consider the work be a promising, well-motivated, and overall safe/robust application of ML for collider data analyses. However the presentation of the work feels unclear in some places and incomplete/inaccurate in others (see requested changes below). Additionally, the paper would probably benefit from a section near the beginning that provides an overview, to help contextualize the contents of the various sections, which currently feel disorganized.

I would recommend publication of the paper in this journal, if the presentation is improved.

Requested changes

1) Page 3: It is not clear what Eqns 1 and 2 mean, or what the variable r stands for. I assume the intended meaning of eqn 1 is that xreco is sampled from p(xreco | xpart), and operationally xreco is computed using xpart and some standard-normally distributed random variables r. If this is the case, it might just be easier to explain this in words instead of equations. 2) Page 3: "guarantee a statistically correct and calibrated output" It is unclear what this means. 3) Page 6: "In our case, we will guarantee the correct descriptions of the intermediate top-mass peaks through the network architecture" It is unclear what this means. Is this about using MMD loss for the intermiate top-mass distributions? 4) A claim is made that iterative cINN in Ref. [62] can be used to reduce model dependence in the unfolding networks. I don't think this is true (explained below). However, one doesn't need a model independent unfolder for the purposes of this paper, so maybe this claim can be taken out. 5) Eqn 3 shows the loss function for mapping to a Gaussian latent space, instead of not a generic latent space. However, this work uses uniform latent spaces as well. The loss function for that case could be provided. 6) In Eqn 5, does k take on different values for different x? If so, it is not obvious from the equation or the text. 7) Section 3.2 is confusing in a number of ways. Given that the approach is to only use the unfolding network as an analysis variable, the first part of section 3.2 feels unnecessary.
In eqn 26, p(alpha | xreco) and p(alpha | xpart) only make sense if there is a prior on alpha. p(xpart | xreco, B) is not defined, since xpart (in the chosen phase space parameterization) doesn't exist for background events. This renders Eqns 27, 28 and "p(xpart | xreco, B) = p(xpart)" meaningless in my opinion. 8) It is unclear how a variable number of jets is handled in section 3.3. The details weren't immediately obvious to me even after looking through Ref [60]. 9) The input and output shapes of the various neural networks could be provided. 10) MMD is discussed only briefly in page 6. It is unclear whether (and for which distributions) MMD is used in the paper. "Its main advantage is that it only affects the target distribution and avoids an unnecessarily large model dependence." The meaning of this statement is unclear. 11) In figure 6 top 2 rows, it is confusing to have three truth distributions. The non-SM truth curves could be labelled differently in the legend. Also, in those plots, it cINN/truth should probably be computed with the SM truth for all three networks, instead of using different truth curves for each network. 12) A description of how the cINN histograms are created could be provided. For instance, to get the central values of the bin counts in a histogram, is only one parton level event computed for each reco-level event? Or is the central bin-count value computed as an average over many samplings?

Some minor suggestions/comments: 1) Page 14: "The conventional approach to complex kinematic correlations..." Is this supposed to be "compute kinematic correlations"? 2) Page 1: Maybe change "The, arguably, most interesting symmetry in the SM is CP" to "Arguably, the most interesting symmetry in the SM is CP"? 3) Page 1: "CP ... potentially realized in an extended Higgs sector". It is not obvious that the authors are referring to CP violation here.

On model independence:
My understanding of Ref. [62] is as follows. Let's say we have a simulation model A for p(xpart) and another simulation model B for p(xreco | xpart), and let's say model B could be wrong/mismodeled. Then assuming that model A is correct (which is fair to do in control regions only), iterative cINNs can learn an unfolder, which corrects for the mismodeling in B using the experimental data.
In this paper, model dependence is used to mean dependence on model A. I don't think cINNs have been demonstrated to reduce dependence on A. In fact, an unfolder can be completely independent of model A only if the map from xpart to xreco is invertible (i.e., for any given xreco there exists only one possible xpart). This is typically not true in collider physics.
Also, even if we're considering dependence on model B, I'd say the iterative technique corrects for simulation-model errors, and doesn't induce model independence, although that's arguably just semantics.

  • validity: high
  • significance: high
  • originality: high
  • clarity: good
  • formatting: excellent
  • grammar: excellent

Author:  Rahool Kumar Barman  on 2024-03-23  [id 4377]

(in reply to Report 3 by Prasanth Shyamsundar on 2023-10-14)
Category:
answer to question

We would like to thank the referee for the careful evaluation of our manuscript and for providing detailed and valuable comments. The version of the manuscript that we resubmit addresses the aspects that the report brought to our consideration. Please find the comments from the referee and our answers in the attached file. This file can also be accessed via this link: https://www.dropbox.com/scl/fi/fob1gzmwymi6ds2drfuct/Reply_to_report_3.pdf?rlkey=a4afzrs4y3l6vlhq4dihwwie6&dl=0

Attachment:

Reply_to_report_3.pdf

Prasanth Shyamsundar  on 2024-04-04  [id 4388]

(in reply to Rahool Kumar Barman on 2024-03-23 [id 4377])

Warnings issued while processing user-supplied markup:

  • Inconsistency: plain/Markdown and reStructuredText syntaxes are mixed. Markdown will be used.
    Add "#coerce:reST" or "#coerce:plain" as the first line of your text to force reStructuredText or no markup.
    You may also contact the helpdesk if the formatting is incorrect and you are unable to edit your text.

I would like to to thank the authors for addressing/answering my comments and questions thoroughly. I recommend the paper for publication, with only a minor modification (typo) described below.

Regarding point 4 of the original report:
I was completely wrong in my interpretation of iterative cINNs. Thank you for pointing it out, and sorry you had to deal with the comment.

Regarding point 7 of the original report:
I am not fully convinced by the response, but that's okay (it's one out of 12 comments). For what it's worth, I've added a response below, but feel free to ignore it. Also, eq 26 has a typo: $dx$ should be $d x_\mathrm{part}$.

Cheers, Prasanth

PS: Also, it seems like I've missed the deadline for subitting a report on the revision, so it's all moot anyway.

a) When doing a Bayesian estimation of $\alpha$, one would be interested in $p(\alpha~|~D_\mathrm{reco})$, where $D_\mathrm{reco} = (x^{(1)}\mathrm{reco},\dots,x^{(n)}\mathrm{reco})$ is the full reco-level dataset, not just a single datapoint $x_\mathrm{reco}$. Extracting $p(\alpha~|~D_\mathrm{reco})$ from the different $p(\alpha~|~x^{(i)}_\mathrm{reco})$-s alone is a fairly non-trivial task, I think. But let's ignore this and say we're just interested in $p(\alpha~|~x_\mathrm{reco})$ for a given reco-level event.

b) If the goal is to do a Bayesian posterior estimation of $\alpha$ using eq 26, with $p(\alpha)$ as the prior, then unfolding term $p(x_\mathrm{part}~|~x_\mathrm{reco})$ in eq 26 should be learned using data generated with that prior. Attempts to reduce prior dependence (like with iterative cINNs) would undermine task of producing a Bayesian posterior, no?

c) I generally agree with the intuition in this statement: "What Eq. (28) shows, is that we can limit our unfolding model to extracting $p(x_\mathrm{part}|x_\mathrm{reco}, S)$ and still include background events into our analysis later, without changing our model." However the equations that lead to this don't sit right with me. Conditional probabilities have a concrete definition: If $X$ and $Y$ are two simultaneously occuring variables (i.e., they have a joint distribution), then $p(X~|~Y) = p(X, Y)/p(Y)$. If we define $x_\mathrm{part}$ to be a parton level signal event, then $x_\mathrm{part}$ and a background $x_\mathrm{reco}$ do not simultaneously occur. So, to me it is unclear how to interpret their conditional probabilities. I roughly agree with the intuition in the statement "However, the detector-level measurement of a background event gives us no information about the probability of a signal event on parton-level," but conditional probabilities is not the correct outlet for expressing it, in my opinion.

For what it's worth, here's an alternative derivation of your main message in the first part of section 3.2:

For a given event (no restrictions on background or signal), let $x_\mathrm{part}$ and $x_\mathrm{reco}$ be its parton- and reco-level event-description. For a given event, let $S$ and $B$ be the "events" (à la probability-theory) of it coming from signal and background, respectively. Let's assume that $\alpha$ doesn't affect the signal strength, so $p(\alpha~|~S) = p(\alpha~|~B) = p(\alpha)$ (which is the prior). We can begin by writing

$$p(\alpha~|~x_\mathrm{reco}) = p(\alpha~|~x_\mathrm{reco}, B)~p(B~|~x_\mathrm{reco}) + p(\alpha~|~x_\mathrm{reco}, S)~p(S~|~x_\mathrm{reco})$$
Since $\alpha$ doesn't affect signal strength and background events don't provide any information on $\alpha$, $p(\alpha~|~x_\mathrm{reco},B) = p(\alpha)$ . Using this in the first term and introducing $x_\mathrm{part}$ into the second term:
$$p(\alpha~|~x_\mathrm{reco}) = p(\alpha)~p(B~|~x_\mathrm{reco}) + \left[\int d x_\mathrm{part}~p(\alpha~|~x_\mathrm{part}, x_\mathrm{reco}, S)~p(x_\mathrm{part}~|~x_\mathrm{reco}, S)\right]~p(S~|~x_\mathrm{reco})$$
Since $\alpha$ affects $x_\mathrm{reco}$ only through $x_\mathrm{part}$,
$$p(\alpha~|~x_\mathrm{reco}) = p(\alpha)~p(B~|~x_\mathrm{reco}) + \left[\int d x_\mathrm{part}~p(\alpha~|~x_\mathrm{part}, S)~p(x_\mathrm{part}~|~x_\mathrm{reco}, S)\right]~p(S~|~x_\mathrm{reco})$$
From here it can be seen that $p(x_\mathrm{part}~|~x_\mathrm{reco},S)$ is sufficient for estimating the LHS and $p(x_\mathrm{part}~|~x_\mathrm{reco},B)$ isn't needed. Additionally, since $\alpha$ doesn't affect the signal strength, one can write
$$p(\alpha) = p(\alpha~|~S) = \int dx_\mathrm{part}~p(\alpha~|~x_\mathrm{part},S)~p(x_\mathrm{part}~|~S)$$
Plugging this into the previous equation
$$p(\alpha~|~x_\mathrm{reco}) = \int dx_\mathrm{part}~p(\alpha~|~x_\mathrm{part}, S)\Big[p(x_\mathrm{part}~|~S)~p(B~|~x_\mathrm{reco}) + p(x_\mathrm{part}~|~x_\mathrm{reco},S)~p(S~|~x_\mathrm{reco})\Big]$$
This is roughly the same as the result one would get by plugging eq 28 into eq 26 (with some extra conditioning on $S$ in some terms), even though I don't quite agree with eq 28 itself.

---

## Editorial Decision

resubmitted